# Parallelizing Linear Transformers with the Delta Rule over Sequence Length

**Songlin Yang**◇ **Bailin Wang**◇ **Yu Zhang**† **Yikang Shen**‡ **Yoon Kim**◇

◇Massachusetts Institute of Technology †Soochow University ‡MIT-IBM Watson AI Lab

yangsl66@mit.edu

## Abstract

Transformers with linear attention (i.e., linear transformers) and state-space models have recently been suggested as a viable linear-time alternative to transformers with softmax attention. However, these models still underperform transformers especially on tasks that require in-context retrieval. While more expressive variants of linear transformers which replace the additive update in linear transformers with the delta rule [DeltaNet; 101] have been found to be more effective at associative recall, existing algorithms for training such models do not parallelize over sequence length and are thus inefficient to train on modern hardware. This work describes a hardware-efficient algorithm for training linear transformers with the delta rule, which exploits a memory-efficient representation for computing products of Householder matrices [11]. This algorithm allows us to scale up DeltaNet to standard language modeling settings. We train a 1.3B model for 100B tokens and find that it outperforms recent linear-time baselines such as Mamba [31] and GLA [124] in terms of perplexity and zero-shot performance on downstream tasks. We also experiment with two hybrid models which combine DeltaNet layers with (1) sliding-window attention layers every other layer or (2) two global attention layers, and find that these hybrids outperform strong transformer baselines.

## 1 Introduction

The attention mechanism [8, 116] has been shown to be an important primitive for accurate sequence modeling. Attention is moreover efficient during training as it is rich in matrix multiplications and can thus take advantage of highly parallel processing capabilities and specialized accelerators on modern GPUs. However, the complexity of attention is quadratic in sequence length, and hence it is a fundamentally expensive primitive. And while recent techniques have made it possible to scale attention to longer sequences through hardware-aware restructuring of the intermediate computations [20, 18, 59, 14], these methods still require storing the key/value vectors of previous elements, and this "KV cache" (whose size grows linearly) can be unwieldy to manage for long sequences.

Linear attention transformers [48] replace the exponential kernel in softmax attention with a dot-product over (possibly transformed) key and query vectors. This makes it possible to formulate linear attention as a linear RNN with matrix-valued hidden states, thus obviating the need for a KV cache and enabling constant-memory inference. While initial variants of linear attention generally underperformed softmax attention on language modeling, gated variants of linear attention which incorporate a data-dependent gating factor have recently been shown to be competitive against strong transformer baselines [124, 92, 9, 79]. These gated linear transformers, along with time-varying state space models such as Mamba [31, 19] (which can be reparameterized as a gated linear transformer [124]), have been suggested as a potential alternative to ordinary transformers. However, despite

---

The parallel DeltaNet layer is made available as part of the FLASHLINEARATTENTION library [124, 123]: https://github.com/fla-org/flash-linear-attention

38th Conference on Neural Information Processing Systems (NeurIPS 2024).

the competitive language modeling performance, these models have been shown to underperform transformers on recall-intensive tasks [6, 7], which is important for many practical downstream tasks of interest (e.g., in retrieval-augmented generation [53]).

To enhance associative recall over long contexts, Schlag et al. [101] propose DeltaNet, a variant of a linear transformer which uses a delta rule-like update [121] to retrieve and update a value vector that is associated with the current key. DeltaNet was found to be effective on synthetic tasks and small scale language modeling/machine translation. However, the original work used a sequential algorithm that did not parallelize across sequence length, thus resulting in hardware-inefficient training, and it has not been clear how to scale DeltaNet to larger models and datasets.

This work describes a hardware-efficient training algorithm for DeltaNets which parallelizes the forward/backward passes across sequence length. We reparameterize the DeltaNet as a matrix-valued RNN whose recurrence is given by a generalized Householder transformation. This reparameterization enables the use of the compact WY representation [11] for products of Householder matrices, eliminating the need to materialize the hidden states of matrix size at each time step during parallel training, which would otherwise result in high I/O costs. The memory-efficient representation makes it possible to straightforwardly extend the chunkwise parallel strategy for training linear attention models [34, 108, 124] to the DeltaNet case. We scale DeltaNets to moderate-scale language modeling benchmarks (1.3B models trained on 100B tokens), where DeltaNet is found to obtain better language modeling and zero-shot downstream task performance than strong linear recurrent models such as Mamba [31] and GLA [124]. For in-context retrieval and learning evaluation, we evaluate DeltaNet on synthetic and real benchmarks [4, 2, 84, 6], where it is again found to perform well against linear recurrent baselines. Finally, we experiment with a hybrid approach where we combine DeltaNet layers with sliding attention layers or global attention layers, and find that these hybrid models can improve upon ordinary transformers, as well as the pure DeltaNet transformer.

## 2 Background

### 2.1 Linear Transformer: Transformers with Linear Attention

Given a sequence of $d$-dimensional input vectors $\boldsymbol{x}_1, \ldots, \boldsymbol{x}_L$, transformers use the softmax attention mechanism to attend over the entire past,

$$\boldsymbol{q}_t, \ \boldsymbol{k}_t, \ \boldsymbol{v}_t = \boldsymbol{W}_Q \boldsymbol{x}_t, \boldsymbol{W}_K \boldsymbol{x}_t, \boldsymbol{W}_V \boldsymbol{x}_t, \qquad \boldsymbol{o}_t = \sum_{i=1}^{t} \frac{\exp(\boldsymbol{k}_i^\top \boldsymbol{q}_t)}{\sum_{j=1}^{t} \exp(\boldsymbol{k}_j^\top \boldsymbol{q}_t)} \boldsymbol{v}_i,$$

where $\mathbf{W}_Q, \mathbf{W}_K, \mathbf{W}_V \in \mathbb{R}^{d \times d}, \boldsymbol{q}_t, \boldsymbol{k}_t, \boldsymbol{v}_t, \boldsymbol{o}_t \in \mathbb{R}^d$. (Here we assume a single attention head for simplicity). Linear attention [48] replaces the exponential kernel $\exp(\boldsymbol{k}_i^\top \boldsymbol{q}_t)$ with the dot-product $\phi(\boldsymbol{k}_i)^\top \phi(\boldsymbol{q}_t)$ where $\phi : \mathbb{R}^d \to \mathbb{R}^n$ is a feature map. This makes it possible to rearrange computations to represent linear attention as a linear RNN with matrix-valued hidden states,

$$\boldsymbol{o}_t = \sum_{i=1}^{t} \frac{\phi(\boldsymbol{k}_i)^\top \phi(\boldsymbol{q}_t)}{\sum_{j=1}^{t} \phi(\boldsymbol{k}_j)^\top \phi(\boldsymbol{q}_t)} \boldsymbol{v}_i = \frac{\left(\sum_{i=1}^{t} \boldsymbol{v}_i \phi(\boldsymbol{k}_i)^\top\right) \phi(\boldsymbol{q}_t)}{\left(\sum_{j=1}^{t} \phi(\boldsymbol{k}_j)^\top\right) \phi(\boldsymbol{q}_t)} = \frac{\mathbf{S}_t \phi(\boldsymbol{q}_t)}{\mathbf{z}_t^\top \phi(\boldsymbol{q}_t)},$$

where $\mathbf{S}_t = \sum_{i=1}^{t} \boldsymbol{v}_i \phi(\boldsymbol{k}_i)^\top \in \mathbb{R}^{d \times n}$ and $\mathbf{z}_t = \sum_{i=1}^{t} \phi(\boldsymbol{k}_i) \in \mathbb{R}^n$. If we allow $n$ to go to infinity, linear attention can use feature maps associated with polynomial kernels to compute a polynomial approximation to the exponential kernel as a dot product, and can thus approximate softmax attention arbitrarily well [6]. The denominator $\mathbf{z}_t^\top \phi(\boldsymbol{q}_t) \in \mathbb{R}$ can result in numerical instabilities [86] and is removed in recent works [101, 63]. It is also common to use the identity mapping for $\phi$ [63, 108], which results in the following simplified linear transformer: $\mathbf{S}_t = \mathbf{S}_{t-1} + \boldsymbol{v}_t \boldsymbol{k}_t^\top, \boldsymbol{o}_t = \mathbf{S}_t \boldsymbol{q}_t$.

**Efficient training.** Let $\mathbf{Q}, \mathbf{K}, \mathbf{V} \in \mathbb{R}^{L \times d}$ be the stacked query, key, value vectors, e.g., $\mathbf{Q}_i = \boldsymbol{q}_i$. We can then compute the output $\mathbf{O} \in \mathbb{R}^{L \times d}$ in parallel via $\mathbf{O} = \left(\mathbf{Q}\mathbf{K}^\top \odot \mathbf{M}_L\right) \mathbf{V}$, where $\mathbf{M}_L \in \mathbb{R}^{L \times L}$ is the causal mask. This fully "parallel form" and the above "recurrent form" have different FLOPs and parallelization tradeoffs. The parallel form takes $O(L^2 d + L d^2)$ and thus requires more FLOPs than the recurrent form, which takes $O(L d^2)$. However, the parallel form is often much faster in practice for moderate-length sequences as it can be done in $O(1)$ steps. This sequence-level parallellism also enables high GPU occupancy. The recurrent form requires fewer

FLOPs but cannot be parallelized across sequence length[1] and the elementwise operations involved in recurrence moreover cannot make use of specialized matmul accelerators (e.g., tensor cores).

**Chunkwise parallel form.** The chunkwise parallel form [34, 108, 124] strikes a balance between the parallel and recurrent forms, allowing for fewer FLOPs than the parallel form and more sequence-level parallelism than the recurrent form. Concretely, suppose the query/key/value vectors are split into $\frac{L}{C}$ chunks where each chunk is of length $C$. Let $\mathbf{Q}_{[t]} \in \mathbb{R}^{C \times d}$ be all the query vectors for chunk $t$, and let $\boldsymbol{q}_{[t]}^i = \boldsymbol{q}_{tC+i}$ be the $i$-th query vector within the $t$'th chunk; the key/value chunks are defined similarly. Note that $t \in [0, L/C)$, $i \in [1, C]$. The state matrices are also re-indexed such that $\mathbf{S}_{[t]}^i = \mathbf{S}_{tC+i}$, and we additionally define $\mathbf{S}_{[t]}^0 = \mathbf{S}_{[t-1]}^C$, i.e., the initial state of a chunk is the last state of the previous chunk. We can then obtain the following identity for the hidden state and output vector for the $r$-th element within the $t$-th chunk,

$$\mathbf{S}_{[t]}^r = \mathbf{S}_{[t]}^0 + \sum_{i=1}^{r} \boldsymbol{v}_{[t]}^i \boldsymbol{k}_{[t]}^{i\top}, \qquad \boldsymbol{o}_{[t]}^r = \mathbf{S}_{[t]}^0 \boldsymbol{q}_{[t]}^r + \sum_{i=1}^{r} \boldsymbol{v}_{[t]}^i \left( \boldsymbol{k}_{[t]}^{i\top} \boldsymbol{q}_{[t]}^r \right).$$

By further rewriting the intra-chunk computation based on the parallel form, we obtain following,

$$\mathbf{S}_{[t+1]} = \mathbf{S}_{[t]} + \mathbf{V}_{[t]}^\top \mathbf{K}_{[t]} \qquad\qquad \in \mathbb{R}^{d \times d}, \qquad (1)$$

$$\mathbf{O}_{[t]} = \mathbf{Q}_{[t]} \mathbf{S}_{[t]}^\top + \left( \mathbf{Q}_{[t]} \mathbf{K}_{[t]}^\top \odot \mathbf{M}_C \right) \mathbf{V}_{[t]} \qquad \in \mathbb{R}^{C \times d} \qquad (2)$$

where we let $\mathbf{S}_{[t]} = \mathbf{S}_{[t]}^0$ to reduce notational clutter. With this "chunkwise parallel form", information is propagated chunk-to-chunk through $\mathbf{S}_{[t]}$, and the intra-chunk states $\mathbf{S}_{[t]}^i$ for $i \in [1, C]$ need not be materialized, thus saving memory.

The complexity of the chunkwise parallel form is $O(LCd + Ld^2)$, and the number of steps (without chunk-level parallel scan) is $O(\frac{L}{C})$. Hence, $C = L$ recovers the fully parallel form and $C = 1$ recovers the recurrent form. The chunkwise parallel form allows us to interpolate between the two forms, in essence trading off the number of sequential computations against sequence-level parallelism. In practice $C$ is set to a small constant (usually 64 or 128), allowing for subquadratic training. This chunkwise form enables practical speed-ups against parallel-form-only softmax attention even on moderate-length sequences, as demonstrated by FLASHLINEARATTENTION [124, 123]

### 2.2 DeltaNet: Linear Transformers with the Delta Update Rule

The above linear transformer employs a simple linear recurrence: $\mathbf{S}_t = \mathbf{S}_{t-1} + \boldsymbol{v}_t \boldsymbol{k}_t^\top$. This can be seen as additively updating the memory $\mathbf{S}_{t-1}$ with new key-value associations at each time step. However, a purely additive update rule makes it difficult to deallocate past key-value associations, eventually leading to key "collisions" when $L > d$, as pointed out by Schlag et al. [101]. A model should ideally learn to remove less important key-value associations to make room for new ones, and this removal should depend on the interaction between the new key and the memory content.

From a fast weight programming [102] perspective, the recurrent hidden state of linear attention is the fast weight mapping the input $\boldsymbol{q}_t$ to output $\boldsymbol{o}_t$ with a Hessian-like update rule, which has limited memory capacity [68]. DeltaNet, on the other hand, uses the delta update rule or the Widrow-Hoff learning rule [121] for fast weight update: $\mathbf{S}_t = \mathbf{S}_{t-1} - \beta_t(\mathbf{S}_{t-1}\boldsymbol{k}_t - \boldsymbol{v}_t)\boldsymbol{k}_t^\top$, where $\beta_t$ is the learning rate, $\mathbf{S}_{t-1}\boldsymbol{k}_t$ represents the current prediction, $\boldsymbol{v}_t$ is the target value. The method derives its name from the core principle of updating weights based on the "delta" (difference) between the prediction $\mathbf{S}_{t-1}\boldsymbol{k}_t$ and the target $\boldsymbol{v}_t$, which has been shown better memory capacity [28, 85, 56, 101]. This process can also be regarded as optimizing an online regression loss using a single step of SGD,[2]

$$\mathcal{L}_t(\mathbf{S}) = \frac{1}{2}\|\mathbf{S}\boldsymbol{k}_t - \boldsymbol{v}_t\|^2, \quad \mathbf{S}_t = \mathbf{S}_{t-1} - \beta_t \nabla_{\mathbf{S}_{t-1}} \mathcal{L}_t(\mathbf{S}_{t-1}) = \mathbf{S}_{t-1} - \beta_t(\mathbf{S}_{t-1}\boldsymbol{k}_t - \boldsymbol{v}_t)\boldsymbol{k}_t^\top,$$

which has been discussed in several recent works [110, 58, 125, 10]. In contrast, linear transformers

---

[1]It is possible in theory to use parallel scan [13] to parallelize the recurrent form, which would enable the computations to be performed in $O(\log L)$ steps and $O(Ld^2)$ FLOPs. However, this approach requires materializing the 2D hidden state for each time step, which would incur significant memory I/O cost unless the state size is small enough such that materialization can happen in faster memory (i.e., as in Mamba [31]).

[2]This perspective was discussed as early as Widrow et al. [121], which later became known as the Least Mean Squares (LMS) algorithm and has found widespread applications in signal processing [97].

can be regarded as optimizing an online linear (negative inner-product) loss $\mathcal{L}_t = -\langle \mathbf{S}\boldsymbol{k}_t, \boldsymbol{v}_t \rangle$.[3]

An alternative interpretation for DeltaNet is from the perspective of key-value retrieval. It first retrieves the old value using the current key, $\boldsymbol{v}_t^{\text{old}} = \mathbf{S}_{t-1}\boldsymbol{k}_t$. It then obtains a new value $\boldsymbol{v}_t^{\text{new}}$ by interpolating between the old value and the current value $\boldsymbol{v}_t$, which replaces $\boldsymbol{v}_t^{\text{old}}$ in the memory:

$$\boldsymbol{v}_t^{\text{new}} = \beta_t \boldsymbol{v}_t + (1 - \beta_t)\, \boldsymbol{v}_t^{\text{old}}, \qquad \mathbf{S}_t = \mathbf{S}_{t-1} \underbrace{-\boldsymbol{v}_t^{\text{old}}\boldsymbol{k}_t^{\top}}_{\text{remove}} \underbrace{+\boldsymbol{v}_t^{\text{new}}\boldsymbol{k}_t^{\top}}_{\text{write}}$$

Here $\beta_t = \sigma(\mathbf{W}_\beta \boldsymbol{x}_t) \in (0, 1)$ is a soft "writing strength": when $\beta_t = 1$, the old value is completely removed and $\boldsymbol{v}_t^{\text{new}} = \boldsymbol{v}_t$; when $\beta_t = 0$, the memory remains unmodified and we have $\mathbf{S}_t = \mathbf{S}_{t-1}$. The output computation is the same as vanilla linear attention, i.e., $\boldsymbol{o}_t = \mathbf{S}_t \boldsymbol{q}_t$. The complexity of this recurrent form is the same as that of vanilla linear attention, i.e., $\mathcal{O}(Ld^2)$.

Schlag et al. [101] demonstrate that DeltaNet outperforms ordinary linear transformers on small-scale language modeling and synthetic in-context retrieval tasks. However, their training algorithm, an extension of the memory-efficient recurrent implementation for linear Transformers [48, §3.3.1], is strictly sequential and thus hardware inefficient, as discussed in [124, §3.2], motivating us to derive an equivalent chunkwise algorithm to train DeltaNet at scale, as introduced below.

## 3  Parallelizing DeltaNet Across the Sequence Dimension

### 3.1  A Memory-efficient Reparameterization

We first observe that $\mathbf{S}_t$ admits a purely additive representation of the form $\mathbf{S}_t = \sum_{i=1}^{t} \boldsymbol{u}_i \boldsymbol{k}_i^{\top}$ for $\boldsymbol{u}_i, \boldsymbol{k}_i \in \mathbb{R}^d$, since we can simply set $\boldsymbol{u}_i = \boldsymbol{v}_i^{\text{new}} - \boldsymbol{v}_i^{\text{old}} = \beta_i(\boldsymbol{v}_i - \boldsymbol{v}_i^{\text{old}})$. Recall from §2.1 that simple linear attention has the form $\mathbf{S}_t = \sum_{i=1}^{t} \boldsymbol{v}_i \boldsymbol{k}_i^{\top}$. Thus, DeltaNet simply replaces the value vector $\boldsymbol{v}_i$ in linear attention with the "pseudo" value vector $\boldsymbol{u}_i$. Once the $\boldsymbol{u}_i$'s have been constructed, the rest of computation can proceed as in ordinary linear attention, i.e., $\mathbf{O} = (\mathbf{Q}\mathbf{K}^{\top} \odot \mathbf{M})\,\mathbf{U}$ where $\mathbf{U} \in \mathbb{R}^{L \times d}$ is the row-wise concatenation of the $\boldsymbol{u}_i$ vectors.

However, computing $\boldsymbol{u}_t$ naïvely requires explicitly materializing $\mathbf{S}_{t-1}$ to compute $\boldsymbol{v}_t^{\text{old}}$, which would require $\mathcal{O}(d^2)$ memory. We now show that we can obtain the $\boldsymbol{u}_t$'s *without* explicitly materializing $\mathbf{S}_{t-1}$ in $\mathcal{O}(d)$ memory. Our simple proof (by induction) relies on an application of the WY representation for products of Householder matrices [11]. The base case is clear since we have $\mathbf{S}_1 = \beta_1 \boldsymbol{v}_1 \boldsymbol{k}_1^{\top}$, so $\boldsymbol{u}_1 = \beta_1 \boldsymbol{v}_1$. For the inductive step, we first observe that the DeltaNet update is given by,

$$\mathbf{S}_t = \mathbf{S}_{t-1} - \boldsymbol{v}_t^{\text{old}}\boldsymbol{k}_t^{\top} + \boldsymbol{v}_t^{\text{new}}\boldsymbol{k}_t^{\top} = \mathbf{S}_{t-1} - \beta_t \left(\mathbf{S}_{t-1}\boldsymbol{k}_t\right)\boldsymbol{k}_t^{\top} + \beta_t \boldsymbol{v}_t \boldsymbol{k}_t^{\top} = \mathbf{S}_{t-1}(\mathbf{I} - \beta_t \boldsymbol{k}_t \boldsymbol{k}_t^{\top}) + \beta_t \boldsymbol{v}_t \boldsymbol{k}_t^{\top},$$

which can be seen as applying a generalized Householder transformation (i.e., matmul with an identity plus rank-one matrix) to the previous state. The inductive step is then given by,

$$\mathbf{S}_t = \mathbf{S}_{t-1}(\mathbf{I} - \beta_t \boldsymbol{k}_t \boldsymbol{k}_t^{\top}) + \beta_t \boldsymbol{v}_t \boldsymbol{k}_t^{\top} = \sum_{i=1}^{t-1} \boldsymbol{u}_i \boldsymbol{k}_i^{\top} + \beta_t \underbrace{\left( \boldsymbol{v}_t - \sum_{i=1}^{t-1} \boldsymbol{u}_i \left( \boldsymbol{k}_i^{\top} \boldsymbol{k}_t \right) \right) \boldsymbol{k}_t^{\top}}_{\text{defined as } \boldsymbol{u}_t} = \sum_{i=1}^{t} \boldsymbol{u}_i \boldsymbol{k}_i^{\top} \quad (3)$$

Note that $\boldsymbol{u}_t$ does not require materializing any of the hidden states and requires $\mathcal{O}(d)$ memory to compute, thus completing the proof. While we have avoided materializing $\mathbf{S}_t$'s, computing $\boldsymbol{u}_t$'s for all $L$ (that is, $\mathbf{U}$) takes $\mathcal{O}(L^2 d)$ and moreover cannot be fully parallelized, unlike in linear attention where we can calculate all the value vectors $\mathbf{V}$ in parallel in $\mathcal{O}(1)$ steps. We now show that the above trick still enables an efficient chunkwise parallel form for DeltaNet.

### 3.2  Chunkwise Parallel Form for DeltaNet

To derive the chunkwise parallel form, we first unroll the recurrence,

$$\mathbf{S}_t = \mathbf{S}_{t-1}(\mathbf{I} - \beta_t \boldsymbol{k}_t \boldsymbol{k}_t^{\top}) + \beta_t \boldsymbol{v}_t \boldsymbol{k}_t^{\top} = \sum_{i=1}^{t} \beta_i(\boldsymbol{v}_i \boldsymbol{k}_i^{\top}) \left( \prod_{j=i+1}^{t} (\mathbf{I} - \beta_j \boldsymbol{k}_j \boldsymbol{k}_j^{\top}) \right). \qquad (4)$$

---

[3]While quadratic loss (used in DeltaNet) provides gradients that scale with error magnitude, offering stronger corrections when predictions are far from the target, the gradient of linear loss remains constant regardless of prediction error. This gradient behavior provides an intuitive explanation for why linear transformers underperform DeltaNet in in-context retrieval tasks.

We then define the following variables: $\mathbf{P}_i^j = \prod_{t=i}^{j}(\mathbf{I} - \beta_t \mathbf{k}_t \mathbf{k}_t^\top) \in \mathbb{R}^{d \times d}$, $\mathbf{H}_i^j = \sum_{t=i}^{j} \beta_t(\mathbf{v}_t \mathbf{k}_t^\top)\mathbf{P}_{t+1}^j \in \mathbb{R}^{d \times d}$, where we let $\mathbf{P}_i^j = \mathbf{I}$ whenever $i > j$. Intuitively, $\mathbf{P}_i^j$ is the "decay factor" to be applied to $\mathbf{S}_i$ for obtaining $\mathbf{S}_j$, and $\mathbf{H}_i^j$ represents the contributions to $\mathbf{S}_j$ starting from token $i$. (Hence $\mathbf{S}_t = \mathbf{H}_1^t$). The chunkwise recurrence can then be written as,

$$\mathbf{S}_{[t]}^r = \mathbf{S}_{[t]}^0 \mathbf{P}_{[t]}^r + \mathbf{H}_{[t]}^r \tag{5}$$

where we define the chunkwise variables $\mathbf{S}_{[t]}^i = \mathbf{S}_{tC+i}$, $\mathbf{P}_{[t]}^r = \mathbf{P}_{tC+1}^{tC+r}$, $\mathbf{H}_{[t]}^r = \mathbf{H}_{tC+1}^{tC+r}$. Here we have $\frac{L}{C}$ chunks of size $C$. The trick is to now efficiently represent the $\mathbf{P}_{[t]}^r, \mathbf{H}_{[t]}^r \in \mathbb{R}^{d \times d}$ matrices using a similar approach described in §3.1, so that these matrices can be stored in $\mathcal{O}(d)$ memory,

$$\mathbf{P}_{[t]}^r = \mathbf{I} - \sum_{i=1}^{r} \mathbf{w}_{[t]}^i \mathbf{k}_{[t]}^{i\top}, \qquad \mathbf{H}_{[t]}^r = \sum_{i=1}^{r} \mathbf{u}_{[t]}^i \mathbf{k}_{[t]}^{i\top} \quad \in \mathbb{R}^{d \times d} \tag{6}$$

$$\mathbf{w}_{[t]}^r = \beta_{[t]}^r \left( \mathbf{k}_{[t]}^r - \sum_{i=1}^{r-1} \mathbf{w}_{[t]}^i (\mathbf{k}_{[t]}^{i\top} \mathbf{k}_{[t]}^r) \right), \quad \mathbf{u}_{[t]}^r = \beta_{[t]}^r \left( \mathbf{v}_{[t]}^r - \sum_{i=1}^{r-1} \mathbf{u}_{[t]}^i (\mathbf{k}_{[t]}^{i\top} \mathbf{k}_{[t]}^r) \right) \quad \in \mathbb{R}^d \tag{7}$$

The derivations for the above can be found in the appendix. Subsequently, based on Eq. 5, we can obtain the chunk-level recurrence for hidden states and outputs as,

$$\mathbf{S}_{[t]}^r = \mathbf{S}_{[t]}^0 - \left( \mathbf{S}_{[t]}^0 \sum_{i=1}^{r} \mathbf{w}_{[t]}^i \mathbf{k}_{[t]}^{i\top} \right) + \sum_{i=1}^{r} \mathbf{u}_{[t]}^i \mathbf{k}_{[t]}^{i\top} = \mathbf{S}_{[t]}^0 + \sum_{i=1}^{r} \left( \mathbf{u}_{[t]}^i - \mathbf{S}_{[t]}^0 \mathbf{w}_{[t]}^i \right) \mathbf{k}_{[t]}^{i\top},$$

$$\mathbf{o}_{[t]}^r = \mathbf{S}_{[t]}^r \mathbf{q}_{[t]}^r = \mathbf{S}_{[t]}^0 \mathbf{q}_{[t]}^r + \sum_{i=1}^{r} \left( \mathbf{u}_{[t]}^i - \mathbf{S}_{[t]}^0 \mathbf{w}_{[t]}^i \right) \left( \mathbf{k}_{[t]}^{i\top} \mathbf{q}_{[t]}^i \right).$$

Letting $\mathbf{S}_{[t]} = \mathbf{S}_{[t]}^0$, the above can be simplified to matrix notations similarly to Eq.1-2,

$$\mathbf{S}_{[t+1]} = \mathbf{S}_{[t]} + \left( \mathbf{U}_{[t]} - \mathbf{W}_{[t]} \mathbf{S}_{[t]}^\top \right)^\top \mathbf{K}_{[t]}, \tag{8}$$

$$\mathbf{O}_{[t]} = \mathbf{Q}_{[t]} \mathbf{S}_{[t]}^\top + (\mathbf{Q}_{[t]} \mathbf{K}_{[t]}^\top \odot \mathbf{M}) \left( \mathbf{U}_{[t]} - \mathbf{W}_{[t]} \mathbf{S}_{[t]}^\top \right) \tag{9}$$

where $\square_{[t]} = \square_{[t]}^{1:C} \in \mathbb{R}^{C \times d}$ for $\square \in \{\mathbf{Q}, \mathbf{K}, \mathbf{V}, \mathbf{O}, \mathbf{U}, \mathbf{W}\}$ defines the chunkwise matrices that are formed from stacking the $\mathbf{q}_t, \mathbf{k}_t, \mathbf{v}_t, \mathbf{o}_t, \mathbf{u}_t, \mathbf{w}_t$ vectors.

**Practical considerations.** In the above, Eq. 7 is fully recurrent and thus cannot use tensor cores written as is. To solve this, we further leverage the *UT transform* [44, 23] (see §B.2 for derivations):

$$\mathbf{T}_{[t]} = \left( \mathbf{I} + \text{tril}(\text{diag}(\beta_{[t]}) \mathbf{K}_{[t]} \mathbf{K}_{[t]}^\top, -1) \right)^{-1} \text{diag}\left( \beta_{[t]} \right) \tag{10}$$

$$\mathbf{W}_{[t]} = \mathbf{T}_{[t]} \mathbf{K}_{[t]}, \qquad\qquad \mathbf{U}_{[t]} = \mathbf{T}_{[t]} \mathbf{V}_{[t]} \tag{11}$$

to rewrite most operations in matmuls. The inverse of lower triangular matrices could be solved efficiently using forward substitution. Once computed, the hidden state updates (Eq. 8) and the output computations (Eq. 9) are largely the same as in vanilla linear attention. We adapt FLASHLINEARATTENTION [124] to implement Eq. 8 and 9 with hidden states recomputed during the backward pass for saving GPU memory. The PyTorch pseudocode for the forward pass is shown in Listing 1.

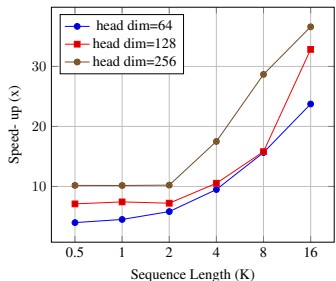

**Figure 1:** Speed-up of the chunkwise parallel form vs. the recurrent form.

**Speed comparison.** We implement both the pure recurrent form[4] and the chunkwise parallel form in Triton [112] and show the speed-ups for various sequence lengths (L) and head dimensions ($d_{\text{head}}$) in the right figure, where the model dimension $d$ is 2048 and we vary batch size and sequence length so that they multily to 16384.[5] Our chunkwise algorithm achieves greater speed-ups as sequence length $L$ and head dimension $d_{\text{head}}$ increase, where the use of sequence-level parallelism (for high GPU occupancy) and tensor core (for fast matmuls) become more important [124, §3].

---

[4]Note that our recurrent kernel is already $2\times$ faster than the original CUDA kernel from Schlag et al. [101].

[5]So far we have been assuming a single head ($d_{\text{head}} = d$) for easier exposition. In practice we use multiple heads where the head dimension $d_{\text{head}}$ is smaller than the model dimension $d$. We thus have $\mathbf{S}_t \in \mathbb{R}^{d \times d_{\text{head}}}$.

**Fully Parallel Form for DeltaNet.**  For completeness, we also discuss the fully parallel form of DeltaNet. While we use the concept of a "pseudo" value, it is possible to avoid modifying values. From Eq. 4, it is straightforward to compute the attention matrix $\mathbf{A}$: $\mathbf{A}_{ij} = \boldsymbol{k}_j^\top \mathbf{P}_{j+1}^i \boldsymbol{q}_i$ if $j \leq i$ and $0$ otherwise. Notably, $\mathbf{A}$ has the matrix form $\mathbf{A} = \left(\mathbf{Q}\mathbf{K}^T \odot \mathbf{M}\right)\mathbf{T}$, obtained by combining Eq. 3 and 11. However, computing $\mathbf{T}$ requires a matrix inverse (Eq. 10), which scales cubically with sequence length without further algorithmic changes. Due to the above we avoid using the fully parallel form for training DeltaNet; however the "attention" matrix derived from this form could be of interest to the interpretability study for RNNs [3, 134].

### 3.3  DeltaNet Transformer

We describe how the DeltaNet layer primitive is used to build up a transformer-like model using standard modules.  We largely follow the LLaMA-architecture [Transformer++, 114] and simply replace the self-attention layer with the DeltaNet layer. We also apply normalization before output projection for stable training [86, 69]. As the additional parameters for computing scalar $\beta_t$ terms are negligible, parameter allocation is roughly the same as in Transformer++, i.e., $4d^2$ for the DeltaNet layer and $8d^2$ for the SwiGLU FFN layer [103].

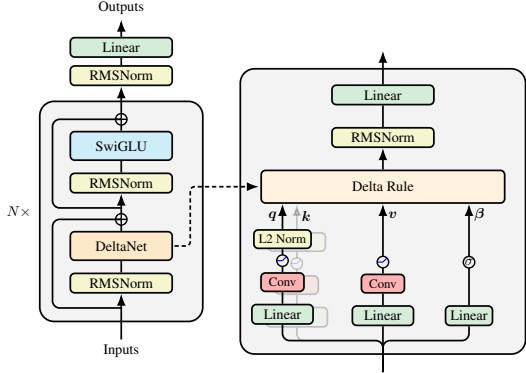

**Figure 2:** An illustration of DeltaNet neural architecture.

**Feature map and normalization.**  Our key/query vectors are given by $\boldsymbol{k}_t = \frac{\text{SiLU}(\mathbf{W}_K\boldsymbol{x}_t)}{\|\text{SiLU}(\mathbf{W}_K\boldsymbol{x}_t)\|_2}$, $\boldsymbol{q}_t = \frac{\text{SiLU}(\mathbf{W}_Q\boldsymbol{x}_t)}{\|\text{SiLU}(\mathbf{W}_Q\boldsymbol{x}_t)\|_2}$. Schlag et al. [101] originally follow Katharopoulos et al. [48] and apply a "ELU$+1$" [17] to nonlineary transform the key/query vectors. We instead use the SiLU activation [24], which was found to perform better [88, 19]. For stability, it is crucial to ensure that the norm of each eigenvalue of the transition matrices does not exceed one. The eigenvalues of $\mathbf{I} - \beta_t\boldsymbol{k}_t\boldsymbol{k}_t^\top$ are 1 with multiplicity $d-1$ and $1 - \beta_t\|\boldsymbol{k}_t\|_2$ with multiplicity 1. Schlag et al. [101] used the $L_1$ norm to normalize query/key vectors, ensuring that $0 \leq 1-\beta_t\|\boldsymbol{k}_t\|_2 \leq 1$. We instead apply $L_2$ normalization, which we found to perform better and offers a more intuitive interpretation: when $\beta_t = 1$, $\mathbf{I} - \boldsymbol{k}_t\boldsymbol{k}_t^\top$ becomes a projection matrix, erasing information in one subspace while preserving the other $d - 1$ subspaces. This is beneficial for retaining information while enabling more *targeted* forgetting.

### 3.4  Hybrid Models

Following recent work on combining subquadratic token-mixing layers with existing neural network primitives [6, 21, 55], we also experiment with hybridizing DeltaNet models.

**Convolutional layers.**  Recent linear recurrent models typically incorporate a lightweight depthwise-separable convolution layer after the query/key/value projections [31, 9, 19]. This "short convolution" layer [83] generalizes the shift SSM [26], and is efficient in both number of parameters and computational cost. We also add a short convolution layer after the query/key/value projections.

**Local sliding window and global attention.**  Linear attention largely uses a content-based addressing mechanism [29] and lacks positional information [129]. Arora et al. [6] also argue that linear attention lacks the ability to perform precise local token shifts and comparisons, thus facing difficulties on retrieval-intensive tasks. Motivated by this, we experiment with two different hybrid architectures that incorporate softmax attention. We first explore *sliding window attention* (SWA) which has been shown to significantly improve linear attention [86, 6, 57, 75]; we follow Griffin [21] and Samba [95] to interleave DeltaNet layers and SWA layers. We also experiment with *global attention*, which has been found to be helpful [51, 35] even if only few of the recurrent layers are replaced with global attention [55]. We follow Fu et al. [26] to replace only two layers with global attention: the second layer and the $(\frac{N}{2} + 1)$-th layer, where $N$ is total number of layers.

## 4  Empirical Study

We compare the DeltaNet against strong baselines in both synthetic and real-world language modeling settings. Our main baselines include: LLaMA-architecture Transformer++ [114]; RetNet [108], a linear attention Transformer with non-data-dependent exponential decay and large head dimension;

GLA [124], a linear attention Transformer with data-dependent decay; and Mamba [31], a selective state-space model with data-dependent decay.

## 4.1 Synthetic Benchmarks

We evaluate on three synthetic benchmarks: Multi-query associative recall [MQAR; 4], Mechanistic Architecture Design [MAD; 84], and in-context language learning [RegBench; 2].

| Model | Compress | Fuzzy Recall | In-Context Recall | Memorize | Noisy Recall | Selective Copy | Average |
|---|---|---|---|---|---|---|---|
| Transformer | 51.6 | 29.8 | 94.1 | 85.2 | 86.8 | 99.6 | **74.5** |
| Hyena | 45.2 | 7.9 | 81.7 | **89.5** | 78.8 | 93.1 | 66.0 |
| Multihead Hyena | 44.8 | 14.4 | 99.0 | 89.4 | 98.6 | 93.0 | 73.2 |
| Mamba | **52.7** | 6.7 | 90.4 | **89.5** | 90.1 | 86.3 | 69.3 |
| GLA | 38.8 | 6.9 | 80.8 | 63.3 | 81.6 | 88.6 | 60.0 |
| DeltaNet | 42.2 | **35.7** | **100** | 52.8 | **100** | **100** | 71.8 |

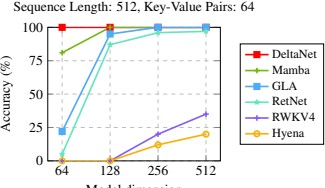

Sequence Length: 512, Key-Value Pairs: 64

**Figure 3:** Results on the synthetic MAD benchmark. Results other than DeltaNet are directly borrowed from Poli et al. [84]. (Multi-head) Hyena, DeltaNet and Mamba make use of convolutions, whereas GLA does not.

**Figure 4:** Accuracy (%) on MQAR.

MQAR evaluates language models' ability to (in-context) recall information within a context when faced with multiple recall queries. We use Arora et al. [4]'s training setting and for DeltaNet we use 2 heads. We do not use convolutions for these experiments. Figure 4 shows that DeltaNet performs perfectly (even without convolution) in the hardest setting and outperforms Mamba (which uses convolutions) in the low-dimension setting. Next, we consider the MAD benchmark [84], a suite of synthetic token manipulation tasks designed to probe capabilities of model architectures. The results are shown in Table 3. Compared with other architectures, including MHA, DeltaNet is better at recalling tasks, especially on Fuzzy Recall as expected, although it somehow struggles on the "Memorize" task. We defer the RegBench results to the §A.2 due to space constraints.

## 4.2 Language Modeling

**Experimental setup.** Following prior work [31, 124], we evaluate on Wikitext perplexity and zero-shot common sense reasoning tasks, including LAMBADA [LMB.; 77], PiQA [12], HellaSwag [Hella.; 127], WinoGrande [Wino.; 99], ARC-easy (ARC-e) and ARC-challenge (Arc-c) [16]. Following Arora et al. [6], we also evaluate the models real-world recall-intensive tasks, including FDA [5], SWDE [60], and SQUAD [93]. Both SWDE and FDA focus on extracting structured information: SWDE from raw HTML to identify semi-structured relationships, and FDA from PDFs to retrieve key-value pairs. SQUAD evaluates language models on reading comprehension by providing a text passage and a related question. See §A.1 for hyperparameter settings.

**Results.** Our main language modeling results are shown in Table 1. Since Mamba uses convolutions by default while GLA does not, we retrain the GLA with convolution, and also train DeltaNet without convolution. For the 1.3B setting we only train the DeltaNet with convolution due to limited compute resources. In general we find that DeltaNet outperforms the strong Mamba/GLA baselines in terms of both perplexity and downstream task performance. For recall-intensive tasks (i.e., SWDE, SQuAD, FDA), we find that under the same state size at the 340M scale, DeltaNet outperforms GLA, confirming the effectiveness of the delta rule. However, at the 1.3B scale, DeltaNet underperforms GLA due to its poorer state size scability (see §5.3), since state size plays an important role in recall-intensive tasks. Finally, we confirm the benefits of hybrid architectures [21, 55]: both the sliding window and global attention hybrids work well, outperforming the strong Transformer++ baselines. We also scale DeltaNet to the 3B parameter scale trained with 1T tokens using the same settings as Shen et al. [104]. The results are shown in Table 5, where 3B DeltaNet slightly underperforms a Transformer architecture trained with the same setting (PowerLM-3B), but outperforms other RNN baselines in the 2B–3B range (though these are trained for a different number of tokens so are not exactly comparable).

**Ablations.** In Table 1 (bottom) we ablate the choice of feature map and normalization. We find that simply replacing the $L_1$-norm with the $L_2$-norm greatly increases performance. For the feature map, we experiment with {ReLU, 1 + ELU, SiLU} and find that SiLU performs the best, consistent with prior work [88].

**Training throughput.** Figure 6 compares the training throughputs of different 1.3B models in different training lengths and batch size settings. The training speed of DeltaNet is close to GLA

| Model | Wiki. ppl ↓ | LMB. ppl ↓ | LMB. acc ↑ | PIQA acc ↑ | Hella. acc_n ↑ | Wino. acc ↑ | ARC-e acc ↑ | ARC-c acc_n ↑ | Avg. | SWDE acc ↑ | SQuAD acc ↑ | FDA acc ↑ | State exp. |
|---|---|---|---|---|---|---|---|---|---|---|---|---|---|
| *340M params / 15B tokens* | | | | | | | | | | | | | |
| Transformer++ | 28.39 | 42.69 | 31.0 | 63.3 | 34.0 | 50.4 | 44.5 | 24.2 | 41.2 | 42.2 | 22.1 | 21.4 | N/A |
| RetNet (*w/o. conv*) | 32.33 | 49.19 | 28.6 | 63.5 | 33.5 | 52.5 | 44.5 | 23.4 | 41.0 | 13.3 | 27.6 | 2.9 | 512x |
| Mamba (*w. conv*) | 28.39 | 39.66 | 30.6 | 65.0 | 35.4 | 50.1 | 46.3 | 23.6 | 41.8 | 12.4 | 23.0 | 2.1 | 64x |
| GLA (*w/o. conv*) | 28.65 | 43.35 | 30.3 | 64.8 | 34.5 | 51.4 | 45.1 | 22.7 | 41.5 | 18.6 | 27.2 | 8.1 | 128x |
| (*w. conv*) | 29.47 | 45.53 | 31.3 | 65.1 | 33.8 | 51.6 | 44.4 | 24.6 | 41.8 | 24.0 | 24.7 | 7.3 | 128x |
| DeltaNet (*w/o. conv*) | 29.08 | 50.87 | 30.0 | 63.6 | 33.6 | 51.7 | 46.0 | 23.0 | 41.3 | 24.6 | 26.9 | 4.5 | 128x |
| DeltaNet (*w. conv*) | 28.24 | 37.37 | 32.1 | 64.8 | 34.3 | 52.2 | 45.8 | 23.5 | 42.1 | 26.4 | 28.9 | 12.8 | 128x |
| + Sliding Attn | 27.06 | 38.17 | 33.4 | 64.0 | 35.3 | 50.9 | 45.9 | 23.2 | 42.1 | 39.3 | 32.5 | 18.8 | N/A |
| + Global Attn (2 layers) | 27.51 | 35.04 | 33.5 | 64.0 | 34.5 | 51.7 | 46.0 | 23.3 | 42.1 | 42.9 | 32.1 | 23.1 | N/A |
| *1.3B params / 100B tokens* | | | | | | | | | | | | | |
| Transformer++ | 16.85 | 13.44 | 48.9 | 70.8 | 49.6 | 53.6 | 56.0 | 26.5 | 50.9 | 66.6 | 31.5 | 27.4 | N/A |
| RetNet (*w/o. conv*) | 18.64 | 17.27 | 43.3 | 70.0 | 47.3 | 52.5 | 54.8 | 25.6 | 48.9 | 42.8 | 34.7 | 14.3 | 512x |
| Mamba (*w. conv*) | 17.06 | 13.89 | 46.2 | 72.2 | 40.1 | 54.1 | 59.0 | 28.2 | 50.0 | 41.4 | 35.2 | 6.2 | 64x |
| GLA (*w/o. conv*) | 17.22 | 14.47 | 46.9 | 71.8 | 49.8 | 53.9 | 57.2 | 26.6 | 51.0 | 50.6 | 42.6 | 19.9 | 256x |
| (*w. conv*) | 17.25 | 14.92 | 46.2 | 70.6 | 49.9 | 53.0 | 55.3 | 27.0 | 50.4 | 52.4 | 37.4 | 22.3 | 256x |
| DeltaNet (*w. conv*) | 16.87 | 12.21 | 48.9 | 71.2 | 50.2 | 53.6 | 57.2 | 28.3 | 51.6 | 49.5 | 37.4 | 17.2 | 128x |
| + Sliding Attn | 16.56 | 11.74 | 49.2 | 71.8 | 51.1 | 52.8 | 58.9 | 28.8 | 52.1 | 53.3 | 43.3 | 22.3 | N/A |
| + Global Attn (2 layers) | 16.55 | 12.40 | 48.8 | 70.8 | 50.7 | 54.2 | 58.4 | 28.1 | 51.8 | 71.0 | 43.0 | 29.8 | N/A |
| *DeltaNet Ablations (340M)* | | | | | | | | | | | | | |
| *w. $L_1$-norm & 1+ELU* | 31.12 | 55.96 | 26.3 | 63.9 | 33.0 | 50.9 | 44.3 | 21.8 | 40.1 | 14.5 | 23.9 | 6.2 | 128x |
| *w. $L_2$-norm & 1+ELU* | 28.03 | 37.62 | 32.2 | 65.7 | 34.7 | 51.8 | 45.4 | 22.5 | 42.1 | 23.8 | 28.6 | 13.1 | 128x |
| *w. $L_2$-norm & ReLU* | 28.75 | 43.53 | 30.2 | 64.0 | 33.9 | 48.9 | 45.6 | 22.8 | 40.9 | 27.2 | 26.7 | 9.0 | 128x |

**Table 1:** Main language modeling results against Transformer++, RetNet [108], Mamba [31], and GLA [124]. All models are trained on the same subset of the SlimPajama dataset with the Mistral tokenizer. The Transformer++, RetNet, Mamba, GLA (*w/o. conv*) results are taking from Yang et al. [124]. For hybrid models, "Sliding Attn" interleaves a sliding window attention every other layer, and "Global Attn" uses full global attention on two layers. The 340M/1.3B models are trained for 15B/100B tokens respectively. All results are obtained through `lm-evaluation-harness` [27]. The last column denotes the expansion ratio of the recurrent state size relative to the product of the number of layers and model dimension (see Zhang et al. [131, App. C]).

| Model | ARC | HellaSwag | OBQA | PIQA | WinoGrande | MMLU | Average |
|---|---|---|---|---|---|---|---|
| Llama-3.2-3B [111] | 59.1 | 73.6 | 43.4 | 77.5 | 69.2 | 54.1 | 62.8 |
| PowerLM-3B [104] | 60.5 | 74.6 | 43.6 | 79.9 | 70.0 | 45.0 | 62.3 |
| DeltaNet-3B | 60.4 | 72.8 | 41.0 | 78.5 | 65.7 | 40.7 | 59.8 |
| RecurrentGemma-2B [30] | 57.0 | 71.1 | 42.0 | 78.2 | 67.6 | 31.8 | 57.9 |
| RWKV-6-3B [79] | 49.5 | 68.6 | 40.6 | 76.8 | 65.4 | 28.4 | 54.9 |
| Mamba-2.7B [31] | 50.3 | 65.3 | 39.4 | 75.8 | 63.1 | 26.1 | 53.3 |

**Figure 5:** Zero-shot model performance across selected benchmarks for 3B models. Llama-3.2-3B and PowerLM-3B are Transformer models, while the others are recurrent models. ARC results are averaged over normalized accuracy across ARC-Easy and ARC-Challenge.

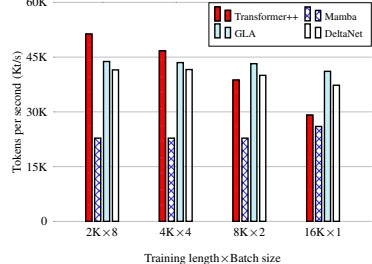

**Figure 6:** Training throughput of 1.3B models on a single H100.

and significantly faster than Mamba. All linear-time models outperform Transformers for longer-sequence training.

## 5 Discussion and Related Work

### 5.1 DeltaNet vs. State Space Models / Linear RNNs

To discuss DeltaNet against existing linear RNNs (including state-space models) we first introduce a general class of associative RNNs with matrix-valued hidden states. Given a matrix-valued hidden state $\mathbf{S}_t \in \mathbb{R}^{d \times n}$ and current input $\boldsymbol{x}_t \in \mathbb{R}^d$, these models have the following form:

$$\mathbf{S}_t = \mathbf{S}_{t-1} \bullet \mathbf{M}_t + \boldsymbol{v}_t \boldsymbol{k}_t^\top, \qquad \text{(recurrence)}$$
$$\boldsymbol{o}_t = \mathbf{S}_t \boldsymbol{q}_t, \qquad \text{(memory read-out)}$$

where $\bullet$ is an associative operator (e.g., Hadamard product, matrix multiplication, etc.). The matrix $\mathbf{M}_t$ and vectors $\boldsymbol{v}_t, \boldsymbol{k}_t, \boldsymbol{q}_t$ are (potentially non-linear) functions of the current input $\boldsymbol{x}_t$.

As is the case in vector-valued linear RNNs [64, 106], the use of an associative operator enables the use of parallel scan [13] to calculate $\mathbf{S}_1, \ldots, \mathbf{S}_L$ in $O(\log L)$ steps and $O(L)$ work (ignoring the terms associated with the associative operation) if the inputs $\boldsymbol{x}_1, \ldots, \boldsymbol{x}_L$ are given (though see our discussion in footnote 1). Hence, as long as the associative operator is not too expensive, training

| Model | Recurrence | Memory read-out |
|---|---|---|
| Linear Attention [48, 47] | $\mathbf{S}_t = \mathbf{S}_{t-1} + \boldsymbol{v}_t \boldsymbol{k}_t^\top$ | $\boldsymbol{o}_t = \mathbf{S}_t \boldsymbol{q}_t$ |
|   + Kernel | $\mathbf{S}_t = \mathbf{S}_{t-1} + \boldsymbol{v}_t \phi(\boldsymbol{k}_t)^\top$ | $\boldsymbol{o}_t = \mathbf{S}_t \phi(\boldsymbol{q}_t)$ |
|   + Normalization | $\mathbf{S}_t = \mathbf{S}_{t-1} + \boldsymbol{v}_t \phi(\boldsymbol{k}_t)^\top, \ \boldsymbol{z}_t = \boldsymbol{z}_{t-1} + \phi(\boldsymbol{k}_t)$ | $\boldsymbol{o}_t = \mathbf{S}_t \phi(\boldsymbol{q}_t)/(\boldsymbol{z}_t^\top \phi(\boldsymbol{q}_t))$ |
| DeltaNet [101] | $\mathbf{S}_t = \mathbf{S}_{t-1}(\mathbf{I} - \beta_t \boldsymbol{k}_t \boldsymbol{k}_t^\top) + \beta_t \boldsymbol{v}_t \boldsymbol{k}_t^\top$ | $\boldsymbol{o}_t = \mathbf{S}_t \boldsymbol{q}_t$ |
| Gated RFA [81] | $\mathbf{S}_t = g_t \mathbf{S}_{t-1} + (1-g_t)\boldsymbol{v}_t \boldsymbol{k}_t^\top, \ \boldsymbol{z}_t = g_t \boldsymbol{z}_{t-1} + (1-g_t)\boldsymbol{k}_t$ | $\boldsymbol{o}_t = \mathbf{S}_t \boldsymbol{q}_t/(\boldsymbol{z}_t^\top \boldsymbol{q}_t)$ |
| S4 [32, 106] | $\mathbf{S}_t = \mathbf{S}_{t-1} \odot \exp(-(\boldsymbol{\alpha}\mathbf{1}^\top) \odot \exp(\boldsymbol{A})) + \boldsymbol{B} \odot (\boldsymbol{v}_t \mathbf{1}^\top)$ | $\boldsymbol{o}_t = (\mathbf{S}_t \odot \boldsymbol{C})\mathbf{1} + \boldsymbol{d} \odot \boldsymbol{v}_t$ |
| ABC [82] | $\mathbf{S}_t^{\boldsymbol{k}} = \mathbf{S}_{t-1}^{\boldsymbol{k}} + \boldsymbol{k}_t \boldsymbol{\phi}_t^\top, \ \mathbf{S}_t^{\boldsymbol{v}} = \mathbf{S}_{t-1}^{\boldsymbol{v}} + \boldsymbol{v}_t \boldsymbol{\phi}_t^\top$ | $\boldsymbol{o}_t = \mathbf{S}_t^{\boldsymbol{v}} \,\mathrm{softmax}\left(\mathbf{S}_t^{\boldsymbol{k}} \boldsymbol{q}_t\right)$ |
| DFW [63] | $\mathbf{S}_t = \mathbf{S}_{t-1} \odot (\boldsymbol{\beta}_t \boldsymbol{\alpha}_t^\top) + \boldsymbol{v}_t \boldsymbol{k}_t^\top$ | $\boldsymbol{o}_t = \mathbf{S}_t \boldsymbol{q}_t$ |
| RetNet [108] | $\mathbf{S}_t = \gamma \mathbf{S}_{t-1} + \boldsymbol{v}_t \boldsymbol{k}_t^\top$ | $\boldsymbol{o}_t = \mathbf{S}_t \boldsymbol{q}_t$ |
| Mamba [31] | $\mathbf{S}_t = \mathbf{S}_{t-1} \odot \exp(-(\boldsymbol{\alpha}_t \mathbf{1}^\top) \odot \exp(\boldsymbol{A})) + (\boldsymbol{\alpha}_t \odot \boldsymbol{v}_t)\boldsymbol{k}_t^\top$ | $\boldsymbol{o}_t = \mathbf{S}_t \boldsymbol{q}_t + \boldsymbol{d} \odot \boldsymbol{v}_t$ |
| GLA [124] | $\mathbf{S}_t = \mathbf{S}_{t-1} \odot (\mathbf{1}\boldsymbol{\alpha}_t^\top) + \boldsymbol{v}_t \boldsymbol{k}_t^\top = \mathbf{S}_{t-1}\mathrm{Diag}(\boldsymbol{\alpha}_t) + \boldsymbol{v}_t \boldsymbol{k}_t^\top$ | $\boldsymbol{o}_t = \mathbf{S}_t \boldsymbol{q}_t$ |
| RWKV-6 [79] | $\mathbf{S}_t = \mathbf{S}_{t-1}\mathrm{Diag}(\boldsymbol{\alpha}_t) + \boldsymbol{v}_t \boldsymbol{k}_t^\top$ | $\boldsymbol{o}_t = (\mathbf{S}_{t-1} + (\boldsymbol{d} \odot \boldsymbol{v}_t)\boldsymbol{k}_t^\top)\boldsymbol{q}_t$ |
| HGRN-2 [92] | $\mathbf{S}_t = \mathbf{S}_{t-1}\mathrm{Diag}(\boldsymbol{\alpha}_t) + \boldsymbol{v}_t(\mathbf{1} - \boldsymbol{\alpha}_t)^\top$ | $\boldsymbol{o}_t = \mathbf{S}_t \boldsymbol{q}_t$ |
| mLSTM [9] | $\mathbf{S}_t = f_t \mathbf{S}_{t-1} + i_t \boldsymbol{v}_t \boldsymbol{k}_t^\top, \ \boldsymbol{z}_t = f_t \boldsymbol{z}_{t-1} + i_t \boldsymbol{k}_t$ | $\boldsymbol{o}_t = \mathbf{S}_t \boldsymbol{q}_t / \max\{1, |\boldsymbol{z}_t^\top \boldsymbol{q}_t|\}$ |
| Mamba-2 [19] | $\mathbf{S}_t = \gamma_t \mathbf{S}_{t-1} + \boldsymbol{v}_t \boldsymbol{k}_t^\top$ | $\boldsymbol{o}_t = \mathbf{S}_t \boldsymbol{q}_t$ |
| GSA [131] | $\mathbf{S}_t^{\boldsymbol{k}} = \mathbf{S}_{t-1}^{\boldsymbol{k}} \mathrm{Diag}(\boldsymbol{\alpha}_t) + \boldsymbol{k}_t \boldsymbol{\phi}_t^\top, \ \mathbf{S}_t^{\boldsymbol{v}} = \mathbf{S}_{t-1}^{\boldsymbol{v}} \mathrm{Diag}(\boldsymbol{\alpha}_t) + \boldsymbol{v}_t \boldsymbol{\phi}_t^\top$ | $\boldsymbol{o}_t = \mathbf{S}_t^{\boldsymbol{v}} \,\mathrm{softmax}\left(\mathbf{S}_t^{\boldsymbol{k}} \boldsymbol{q}_t\right)$ |
| Gated DeltaNet [125] | $\mathbf{S}_t = \mathbf{S}_{t-1}\left(\alpha_t(\mathbf{I} - \beta_t \boldsymbol{k}_t \boldsymbol{k}_t^\top)\right) + \beta_t \boldsymbol{v}_t \boldsymbol{k}_t^\top$ | $\boldsymbol{o}_t = \mathbf{S}_t \boldsymbol{q}_t$ |

**Table 2:** Overview of recent linear recurrent models that have been proposed and applied to autoregressive language modeling (ordered in rough chronological order). These works make use of a matrix-valued hidden state $\mathbf{S}_t \in \mathbb{R}^{d \times n}$ (or two matrix-valued hidden states $\mathbf{S}_t^{\boldsymbol{k}}, \mathbf{S}_t^{\boldsymbol{v}}$, e.g., [82, 131]) updated through an associative recurrence followed by an outer-product-based addition. Here $\odot$ is the Hadamard product. Some models make use of an additional linear RNN with hidden state vector $\boldsymbol{z}_t$, which used to normalized the query vector $\boldsymbol{q}_t$. Variables with the subscript $t$ (e.g., $\boldsymbol{v}_t, \boldsymbol{\alpha}_t, f_t, \gamma_t$) are (potentially non-linear) functions of the current input $\boldsymbol{x}_t$. Non-time-varying parameters (e.g., $\boldsymbol{A}, \boldsymbol{d}, \gamma$) are denoted without subscripts; these parameters are either learned or set to fixed values. Matrices are denoted with bold upper case letters, vectors with bold lower case, and scalars with italic letters. Many models make use of a kernel $\phi$ (e.g., [101, 81]) but we subsume them into the key/value vectors to reduce notational clutter.

can be efficient. However, parallel scan by itself is not sufficient for training language models at practical scale due to some associative operator's being too expensive. Recent models such as such as Mamba [31] and gated linear attention Transformers [108, 124, 92, 79, 9] thus make use of cheap element-wise recurrence updates, in particular the Hadamard product, i.e., $\bullet = \odot$. See Table 2 for how recent models can be cast into this form.

Standard matrix multiplications (i.e., $\mathbf{S}_{t-1} \bullet \mathbf{M}_t = \mathbf{S}_{t-1}\mathbf{M}_t$) on the other hand can model richer interactions that go beyond elementwise recurrence. Without any structural assumptions on $\mathbf{M}_t$ however, these operations would take $O(dn^2)$ for each update (as opposed to $O(dn)$ for elementwise products), which would be prohibitively expensive. Hence, DeltaNet's use of $\mathbf{M}_t = \mathbf{I} - \beta_t \boldsymbol{k}_t \boldsymbol{k}_t^\top$ can be seen as exploiting structured matrices to efficiently model interactions beyond elementwise recurrences. Our chunkwise algorithm could generalize to a broader class of matrices in the Diagonal-Plus-Low-Rank (DPLR) form $\mathbf{M}_t = \mathbf{D} - \boldsymbol{a}_t \boldsymbol{b}_t^\top$, which has been explored in S4 [32], although their DPLR transition matrices are data-independent. We adopt DeltaNet's parameterization in this work (i.e., $\mathbf{D} = \mathbf{I}, \boldsymbol{a}_t = \beta_t \boldsymbol{k}_t, \boldsymbol{b}_t = \boldsymbol{k}_t$) as we are primarily interested in improving recall (through DeltaNet's key-value update rule) while maintaining parameter efficiency. We leave the exploration of more generalized parameterizations for future work.

### 5.2 Towards a Unifying Framework for Efficient Autoregressive Sequence Transformations

While the above class of models makes it possible to unify recent models, we do not claim that it is the "right" level at which view (autoregressive) sequence transformations of the form $\{\boldsymbol{x}_t\}_{t=1}^L \mapsto \{\boldsymbol{o}_t\}_{t=1}^L$, where $\boldsymbol{o}_t$ cannot depend on any $\boldsymbol{x}_j$ if $j > t$. For example, this framing makes it difficult to (neatly) capture other subquadratic models that have been shown to be effective [126, 49, 98, 83]. An alternative unifying framework might be to view the above sequence transformations as a discretization of a continuous state space model [32, 106, 31], or as a matrix multiplication with a masked structured matrix [76, 87, 46, 19]. What does seem important, however, is that a framework should ideally expose efficient algorithms for training, and the algorithm should be hardware-efficient, which, in the case of modern GPUs, means that it should be rich in matrix multiplications. From this perspective, the state-space duality (SSD) framework recently proposed by Dao and Gu [19], which provides a connection between SSM-based sequence transformations and structured matrix multiplications with a semiseparable matrix, seems a promising candidate. However, this framework may not capture an important class of models, e.g., models where the as-

sociative recurrence involves matrix multiplication with an unstructured matrix, or models that make use of more exotic associative operators (e.g., in Peng et al. [80]).

Finally, we observe that there have been many recent linear-time models that have been proposed which purportedly match or outperform classic transformers. As can be seen in Table 2, the "sequence mixing" component of these works are closely related to one another. However, the way in which the token-mixing primitive is used to build up a transformer-like model varies widely. For example, while most recent works make use of depthwise-separable convolution layers (not shown in Table 2) [31, 9, 19, 15, 125], earlier works generally do not [48, 101, 81]. There are also differences in the parameterizations of the feedforward layers used for the "channel mixing" component. Such variations should be taken into account before declaring a particular model layer superior to another.

### 5.3 Limitations and Future Work

Our work has several limitations. First, in terms of computation, although we propose a new hardware-efficient algorithm, the training speed still lags behind that of GLA. This is due to the overhead caused by modeling state-to-state dependencies as described above, which requires "marginalizing" over the head dimension inside the kernel, similar to the case of softmax attention. However, for GLA since there are no intra-state dependencies (everything is elementwise), and thus it is easy to use tiling to support arbitrary size of head dimension, as implemented in Yang and Zhang [123]. This limitation would potentially limit DeltaNet's memory size, consequently lowering the recall-intensive task performance as we observed in §4.2. However, it may be feasible to adopt block diagonal generalized Householder transition matrices with block sizes fitting GPU SRAM (e.g., 128) while maintaining a overall large head dimension (and thus a large recurrent state size).

We also found that the length generalization of DeltaNet was limited, while GLA and RetNet (and Mamba to an extent) have been found to be able to extrapolate beyond the training length [124]. We speculate that this is because DeltaNet lacks explicit decay factors. This could be improved through incorporating a gating term in the recurrence, as demonstrated in a recent work by Yang et al. [125].

## 6 Related Work

We briefly discuss related work here and give an extended discussion in Appendix C.

Linear transformers can be seen as a type of iterated Hopfield networks [72], and this connection can provide perspectives on the limitations and improvements of linear attention transformers. For example, vanilla linear transformers use a Hebbian-like update rule, which has been shown to have limited memory capacity [68]. Later works in Hopfield networks use higher-order polynomials [22] and exponential kernels [94, 50] to enhance the memory capacity, which is also related to linear attention with polynomial kernels [45, 6, 1]. On the other hand, the delta rule has been shown to have better memory capacity [28, 85, 56, 101]. In this sense, given the fixed size recurrent state, using the delta rule is able to achieve a better frontier of the recall-memory tradeoff curve [6], and has recently been applied to enhance real-world retrieval tasks [74, 96]. Moreover, it outperforms the additive rule used in vanilla linear transformers across multiple domains [101, 38, 40, 36, 42].

Despite these advantages, Irie et al. [42] revealed theoretical limitations of the delta update rule in terms of expressiveness. Recurrent enhancements of DeltaNet, such as Recurrent DeltaNet [37] and the Modern Self-Referential Weight Matrix [41], and the mesa-layer [117] were proposed and found superior. However, these models extend beyond linear RNNs and cannot be parallelized across sequence length. This suggests a fundamental trade-off between parallelism and expressiveness [70]. How to further enhance DeltaNet without sacrificing parallelism remains an open question, and the hybrid cross-chunk nonlinear and intra-chunk linear strategy used in TTT [110] might provide a suitable middle ground. Finally, we remark that delta rule is closely related to meta or online learning via gradient descent [73, 39], which has been revisited in recent works like Longhorn [58] and TTT [110]. Recently, Titans [10] improves TTT by introducing a momentum and weight decay term.

## 7 Conclusion

We describe an algorithm that parallelizes DeltaNet training across the sequence length dimension, achieving significant speed-ups against existing implementations on modern hardware. This makes it possible to scale up DeltaNet to moderate-scale language modeling settings, where we find that it performs well compared to recent linear-recurrent baselines.

## Acknowledgements

This study was supported by funds from MIT-IBM Watson AI Lab and the MIT-Google Program for Computing Innovation. We are grateful to Mayank Mishra for assistance with training and evaluating the 3B models, to Kazuki Irie for valuable feedback on the draft, and to Simran Arora, Liliang Ren and Eric Alcaide for their insightful discussions. We also thank Michael Poli and Armin Thomas for sharing the raw results from the MAD benchmark experiment, as well as to Fan Zhou for identifying an error in Eq. 10.

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

## A   Experiments Continued

### A.1   Hyperparameters

We used 8 H100 GPUs for 340M and 1.3B language modeling experiments. Each model uses AdamW for optimization, with a peak learning rate of $3 \times 10^{-4}$. The 340M models are trained using 15 billion tokens and a batch size of 0.5M tokens, while the 1.3B models are trained with 100 billion tokens and a batch size of 2M tokens. We use a cosine learning rate schedule, starting with a warm-up phase of 0.5 billion tokens for the 340M models and 1 billion tokens for the 1.3B models. Both configurations have initial and final learning rates set at $3 \times 10^{-5}$. We apply a weight decay of 0.01 and use gradient clipping at a maximum of 1.0. The head dimension of DeltaNet is set to 128, and the kernel size for convolution layers is set at 4.

### A.2   Synthetic tasks

We additional conduct experiments on RegBench [2], a synthetic data set designed to assess the in-context language learning capability of different model architectures. Each input sequence in this benchmark consists of 10 to 20 strings drawn from a distinct language defined by a probabilistic finite automaton (PFA), so that a model needs to infer the underlying language from the context on the fly. During testing, a model is evaluated on pre-

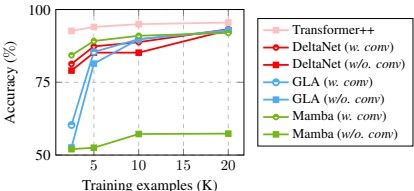

**Figure 7:** Accuracy (%) on RegBench.

dicting the next token of testing sequences generated from held-out PFAs. Here again we find that DeltaNet performs strongly compared to baselines, as shown in Figure 7.

## B   Method Continued

### B.1   WY representation derivation

To reduce notational clutter, we discuss only the first chunk here.

We first show $\mathbf{P}_n = \mathbf{I} - \sum_{t=1}^{n} \boldsymbol{w}_t \boldsymbol{k}_t^{\top}$ by induction,

$$
\begin{aligned}
\mathbf{P}_n &= \prod_{t=1}^{n}(\mathbf{I} - \beta_t \boldsymbol{k}_t \boldsymbol{k}_t^{\top}) \\
&= \mathbf{P}_{n-1}(\mathbf{I} - \beta_n \boldsymbol{k}_n \boldsymbol{k}_n^{\top}) \\
&= (\mathbf{I} - \sum_{t=1}^{n-1} \boldsymbol{w}_t \boldsymbol{k}_t^{\top})(\mathbf{I} - \beta_n \boldsymbol{k}_n \boldsymbol{k}_n^{\top}) \\
&= \mathbf{I} - \sum_{t=1}^{n-1} \boldsymbol{w}_t \boldsymbol{k}_t^{\top} - \beta_n \boldsymbol{k}_n \boldsymbol{k}_n^{\top} + (\sum_{t=1}^{n-1} \boldsymbol{w}_t \boldsymbol{k}_t^{\top})\beta_n \boldsymbol{k}_n \boldsymbol{k}_n^{\top} \\
&= \mathbf{I} - \sum_{t=1}^{n-1} \boldsymbol{w}_t \boldsymbol{k}_t^{\top} - \underbrace{\left(\beta_n \boldsymbol{k}_n - \beta_n \sum_{t=1}^{n-1}\left(\boldsymbol{w}_t(\boldsymbol{k}_t^{\top} \boldsymbol{k}_n)\right)\right)}_{\boldsymbol{w}_n} \boldsymbol{k}_n^{\top} \\
&= \mathbf{I} - \sum_{t=1}^{n} \boldsymbol{w}_t \boldsymbol{k}_t^{\top}
\end{aligned}
$$

Similarly, we show $\mathbf{S}_n = \sum_{t=1}^{n} \boldsymbol{u}_t \boldsymbol{k}_n^{\mathsf{T}}$ by induction,

$$\mathbf{S}_n = \mathbf{S}_{n-1}(\mathbf{I} - \beta_n \boldsymbol{k}_n \boldsymbol{k}_n^{\mathsf{T}}) + \beta_n \boldsymbol{v}_n \boldsymbol{k}_n^{\mathsf{T}}$$

$$= \left( \sum_{t=1}^{n-1} \boldsymbol{u}_t \boldsymbol{k}_t^{\mathsf{T}} \right) (\mathbf{I} - \beta_n \boldsymbol{k}_n \boldsymbol{k}_n^{\mathsf{T}}) + \beta_n \boldsymbol{v}_n \boldsymbol{k}_n^{\mathsf{T}}$$

$$= \sum_{t=1}^{n-1} \boldsymbol{u}_t \boldsymbol{k}_t^{\mathsf{T}} - \left( \sum_{t=1}^{n-1} \boldsymbol{u}_t \boldsymbol{k}_t^{\mathsf{T}} \right) \beta_n \boldsymbol{k}_n \boldsymbol{k}_n^{\mathsf{T}} + \beta_n \boldsymbol{v}_n \boldsymbol{k}_n^{\mathsf{T}}$$

$$= \sum_{t=1}^{n-1} \boldsymbol{u}_t \boldsymbol{k}_t^{\mathsf{T}} + \underbrace{\left( \beta_n \boldsymbol{v}_n - \beta_n \sum_{t=1}^{n-1} \boldsymbol{u}_t \left( \boldsymbol{k}_t^{\mathsf{T}} \boldsymbol{k}_n \right) \right)}_{\boldsymbol{u}_n} \boldsymbol{k}_n^{\mathsf{T}}$$

$$= \sum_{t=1}^{n} \boldsymbol{u}_t \boldsymbol{k}_n^{\mathsf{T}}$$

## B.2 UT transform derivation

Here we provide a detailed derivation of the UT transform used in §3.2. Specifically, we show that the matrix formulation using the UT transform (Equations 10 and 11 in the main text) is equivalent to the recursive update equations (Equation 7).

We begin with the recursive formulation for computing $\boldsymbol{w}_{[t]}^r$ and $\boldsymbol{u}_{[t]}^r$ as given in Equation 7:

$$\boldsymbol{w}_{[t]}^r = \beta_{[t]}^r \left( \boldsymbol{k}_{[t]}^r - \sum_{i=1}^{r-1} \boldsymbol{w}_{[t]}^i (\boldsymbol{k}_{[t]}^{i\mathsf{T}} \boldsymbol{k}_{[t]}^r) \right)$$

$$\boldsymbol{u}_{[t]}^r = \beta_{[t]}^r \left( \boldsymbol{v}_{[t]}^r - \sum_{i=1}^{r-1} \boldsymbol{u}_{[t]}^i (\boldsymbol{k}_{[t]}^{i\mathsf{T}} \boldsymbol{k}_{[t]}^r) \right)$$

Our goal is to show that these recursive updates can be rewritten in matrix form using the UT transform:

$$\mathbf{T}_{[t]} = \left( \mathbf{I} + \mathrm{tril}(\mathrm{diag}(\beta_{[t]}) \mathbf{K}_{[t]} \mathbf{K}_{[t]}^{\mathsf{T}}, -1) \right)^{-1} \mathrm{diag}\left( \beta_{[t]} \right)$$

$$\mathbf{W}_{[t]} = \mathbf{T}_{[t]} \mathbf{K}_{[t]}$$

$$\mathbf{U}_{[t]} = \mathbf{T}_{[t]} \mathbf{V}_{[t]}$$

First, let us consider the computation of $\boldsymbol{w}_{[t]}^r$. For any row $r$, we can write:

$$\mathbf{W}_{[t]}[r, :] = \beta_{[t]}^r \mathbf{K}_{[t]}[r, :] - \beta_{[t]}^r \sum_{i=1}^{r-1} \mathbf{W}_{[t]}[i, :] (\mathbf{K}_{[t]}[i, :] \mathbf{K}_{[t]}[r, :]^{\mathsf{T}})$$

This system of equations can be written in matrix form. Let us define:

$$\mathbf{B}_{[t]} = \mathrm{diag}(\beta_{[t]})$$

$$\mathbf{L}_{[t]} = \mathrm{tril}(\mathbf{B}_{[t]} \mathbf{K}_{[t]} \mathbf{K}_{[t]}^{\mathsf{T}}, -1)$$

Then the system becomes:

$$\mathbf{W}_{[t]} + \mathbf{L}_{[t]} \mathbf{W}_{[t]} = \mathbf{B}_{[t]} \mathbf{K}_{[t]}$$

Thus, we can solve for $\mathbf{W}_{[t]}$:

$$\mathbf{W}_{[t]} = (\mathbf{I} + \mathbf{L}_{[t]})^{-1} \mathbf{B}_{[t]} \mathbf{K}_{[t]} = \mathbf{T}_{[t]} \mathbf{K}_{[t]}$$

where

$$\mathbf{T}_{[t]} = (\mathbf{I} + \mathbf{L}_{[t]})^{-1} \mathbf{B}_{[t]}$$

The same derivation applies for $\mathbf{U}_{[t]}$ by replacing $\mathbf{K}_{[t]}$ with $\mathbf{V}_{[t]}$ in the final step, yielding:

$$\mathbf{U}_{[t]} = \mathbf{T}_{[t]} \mathbf{V}_{[t]} \tag{12}$$

# C  Related Work Continued

**Chunkwise linear attention.**  Hua et al. [34] first proposed chunkwise form for linear attention; however, they used a hybrid linear and nonlinear attention model similar to Munkhdalai et al. [74]. It is possible to adapt their algorithm to compute the *exact* output of the pure linear attention, as shown in Sun et al. [108] and Yang et al. [124]. The chunkwise linear attention algorithm has also been independently discovered in several works [108, 45, 19]. Yang et al. [124] and Qin et al. [91] discuss I/O-aware hardware optimization for chunkwise linear attention and Sun et al. [107] make generalization to multi-node distributed training. Inspired by the chunkwise form, we propose a new algorithm for hardware-efficient DeltaNet training, significantly improving the training efficiency and allowing for large-scale experiments.

**Hybrid models.**  Linear recurrent models - including state-space models [32, 105, 31, 120], gated linear RNNs [65, 90], and linear attention mechanisms [48, 124] - have demonstrated remarkable potential as alternatives to traditional softmax attention across diverse domains [122, 132, 54, 133, 119, 100]. Given their complementary strengths, recent research has increasingly focused on developing hybrid architectures that combine linear recurrent layers with local chunk attention [61, 130, 25, 62, 74] or sliding window attention [130, 6, 21, 95] or global attention [51, 52, 35, 26, 55, 78, 109]. Poli et al. [84] systematically study the scaling law of hybrid models. We similarly show that combining DeltaNet with classic attention is an effective strategy.

**Householder matrices.**  Householder matrices, known for preserving norms, are a type of orthogonal matrix extensively used in machine learning [67, 71, 128, 113, 89, 115]. These matrices allow for efficient computation of inverses and their Jacobian determinant of one, making them particularly suitable for applications in normalizing flows [67, 115]. Notably, Mathiasen et al. [67] and Mathiasen et al. [66] developed a chunkwise fast algorithm for computing the cumulative product of Householder matrices for normalizing flows, leveraging the WY representation. Our approach, while sharing the same high-level concept, tackles a different problem and is arguably more general.

There has also been significant interest in using orthogonal matrices to parameterize the transition matrices of RNNs [71, 43, 118, 33] for mitigating vanishing gradients. Mhammedi et al. [71] use the WY representation to reduce the memory footprint when training nonlinear RNNs with Householder transition matrices.

## D Pseudo code

```python
def chunk_delta_rule_forward(Q, K, V, beta, C):
    '''
    Q/K/V: query, key, value of shape [L, d]
    beta: beta of shape [L]
    C: chunk size
    '''
    # L: sequence length, d: head dimension
    L, d = Q.shape

    # chunking
    Q, K, V = map(lambda x:  x.reshape(-1,C,d), [Q, K, V])
    beta = beta.reshape(-1, C)
    K_beta = K * beta.unsqueeze(-1)
    V_beta = V * beta.unsqueeze(-1)

    # compute eq. 10 with vectorized forword substitution for fast inverse
    T = -(K_beta @ K.t()).tril(-1)
    for i in range(1, C):
        T[i, :i] = T[i, :i] + (T[i, :, None] * T[:, :i]).sum(-2)
    T += torch.eye(C)
    # compute Eq. 11
    W = T @ K_beta
    U = T @ V_beta
    # chunkwise parallel. Eq. 8-9
    S = torch.zeros(d, d)
    O = torch.empty_like(V)
    for i in range(L//C):
        q_i, k_i, w_i = Q[i], K[i], W[i]
        u_i = U[i] - w_i @ S
        o_inter = q_i @ S
        A_i = (q_i @ k_i.t()).tril()
        o_intra = A_i @ u_i
        S += k_i.t() @ u_i
        O[i] = o_intra + o_inter
    return O.reshape(L, d)
```

**Listing 1:** Pytorch-like code snippet of the forward pass of our chunkwise algorithm for training DeltaNet. We omit the dimensions of batch size and number of heads for clarity.

