# OpenReview forum: "Parallelizing Linear Transformers with the Delta Rule over Sequence Length"
_NeurIPS.cc/2024/Conference — NeurIPS 2024 poster_

### Official Review · Reviewer_dem4 · 2024-07-11

**Soundness:** 3
**Presentation:** 3
**Contribution:** 3
**Rating:** 7
**Confidence:** 4

**Summary:**

This paper proposes the Delta Rule method to construct the state updates for Linear Attention. Furthermore, the paper introduces a chunk-wise training approach, allowing the computational cost of training to grow subquadratically with the text length. Experimentally, the paper validates the effectiveness of the model architecture using three synthetic benchmarks: MQAR, MAD, and RegBench. Additionally, the paper uses Common Sense Reasoning and Retrieval tasks in LLM pre-training to verify the model's performance in real-world tasks. The model has been validated at scales ranging from 340M to 1.3B parameters. Furthermore, this paper explores the possibility of combining the Delta Rule with Sliding Window Attention and Global Attention, demonstrating the positive impact of the hybrid architecture on model performance.

**Strengths:**

1. Solid work. The paper provides a good derivation, offering a more general method for state updates in Linear Models.
2. The experiments are comprehensive and effectively demonstrate the validity of the model architecture.

**Weaknesses:**

1. Have you conducted experiments on long context? For example, measuring extrapolation and scenarios akin to "looking for a needle in a haystack"? As a linear model, I would like you to further discuss its capability to generalize to long context.
2. The algorithmic speed of Delta Net increases linearly, but it seems to be slower than GLA. Can you analyze the factors contributing to this?
3. Could you further explain the insights of the Delta Net updates? I understand there are algorithmic differences compared to GLA operators, but what unique benefits do they bring? Is there any theoretical analysis?

**Questions:**

I would like to discuss the following questions with you:

Do you think linear models can fundamentally bridge the gap with transformers in memory-based tasks?

Is there an inherent conflict between the ability to handle long context and the performance of memory-based tasks?

---

> ### Author Rebuttal · Authors · 2024-08-07
>
> Thanks for your review!
>
> ## W1 Long context experiments
> Thanks for your suggestion. Models at 1B scale are currently not powerful enough to provide meaningful results on needle-in-the-haystack style long-range benchmarks. We are currently training models are larger scale (3B parameters), and will run the needle-in-the-haystack tests with these models once they have finished training. (We give the 3B model results, which have been trained on 167B tokens so far, on the other benchmarks. We find that DeltaNet performs comparably to Transformers at this scale.)
>
>
> | Model      | # Tokens  | wikitext PPL | arc-c | arc-e | boolq | hellaswag | openbookqa | piqa  | sciq  | winogrande | average   |
> |------------|--------|-------|-------|-------|-----------|------------|-------|-------|------------|-----------|--------------|
> | 3B-Transformer++   | 167B   | 15.39 | 29.78 | 62.96 | 63.06 | 46.15     | 27.8       | 72.96 | 91.1  | 60.62      | 56.80  |
> | 3B-DeltaNet| 167B   | 15.34 | 29.78 | 63.8  | 65.08 | 46.57     | 26.2       | 74.32 | 88.5  | 58.09      | 56.54   |
>
> ## W2: reason of deltanet being slower than GLA.
> Both DeltaNet and GLA scale linearly with respect to sequence length; however, the recurrent transition matrices in GLA are diagonal matrices, making them amenable to tiling techniques (commonly used to accelerate matrix multiplications on GPUs) because the hidden states are independent of each other. The recurrent transition matrices in DeltaNet are more expressive, modeling state-to-state dependencies and thus requiring "marginalizing" over the entire head dimension, making tiling less amenable, as discussed in the limitations section (lines 301-306).
>
> Additionally, the computation of the WY representation in DeltaNet is more expensive than computing the cumulative product of decays in GLA. From Eq. 7-8, we can see that the DeltaNet chunkwise form involves more matrix multiplications than vanilla linear attention (and also GLA).
>
>
>
>
> ## W3 Intuitive explanation and theoretical justification of delta update rule
>
>
> We found [1] motivating the delta rule intuitively from the perspective of key-value associative memory: the key intuition of the delta rule is to subtract the value associated with the current input key, namely the old value, from the memory, and write a new value that is a linear combination of the old value and the current input value into the memory. This encourages the formation of key-value associations, making retrieval easier.
>
> Regarding theoretical justification, in lines 322-323, we reference several theoretical papers that reveal the superiority of the delta rule in memory capacity in a mathematical way.
>
> [1] Linear Transformers Are Secretly Fast Weight Programmers https://arxiv.org/pdf/2102.11174
>
>
>
> ## Q1 Do you think linear models can fundamentally bridge the gap with transformers in memory-based tasks?
>
>
> We don't think linear models cannot fundamentally bridge the gap with Transformers in memory-based (or recall-intensive) tasks. Several theoretical papers reveal such limitations [1, 2].
>
> Still, we believe the use of the delta rule will push the Pareto frontier of the recall-memory tradeoff curve, a concept that was advocated in [2].
>
> It is possible and promising to replace the subquadratic encoder in YOCO [3] with DeltaNet to fully bridge the gap with Transformers in memory tasks.
>
>
>
>
> [1] RNNs are not Transformers (Yet): The Key Bottleneck on In-context Retrieval. https://arxiv.org/abs/2402.18510
>
> [2] Simple linear attention language models balance the recall-throughput tradeoff https://arxiv.org/abs/2402.18668
>
> [3] You Only Cache Once: Decoder-Decoder Architectures for Language Models https://arxiv.org/abs/2405.05254
>
> ## Q2: Is there an inherent conflict between the ability to handle long context and the performance of memory-based tasks?
>
>
> This is a good question, and one for which the community doesn't have a good answer for yet (in our opinion). Our sense is that memory-based tasks will always require some attention-like mechanism which "retrieves" portions of thecontxt. It is possible that subquadratic-but-superlinear attention mechanisms (e.g., those based on clustering) may enable efficient long-context modeling while still enabling good performance on memory-based tasks.

---

> > ### Comment · Reviewer_dem4 · 2024-08-12
> >
> > Thanks for the response, the authors addressed my concerns, and I have raised my score.

---

### Official Review · Reviewer_pqA2 · 2024-07-13

**Soundness:** 4
**Presentation:** 3
**Contribution:** 3
**Rating:** 8
**Confidence:** 3

**Summary:**

This paper introduces a novel algorithm for the efficient training of DeltaNet Linear Transformers. DeltaNet enhances contextual associative recall using a delta rule-like update but was previously limited by inefficient parallelization in its training algorithm. The work described in this paper presents a hardware-efficient algorithm that leverages the memory-efficient WY representation for computing products of Householder matrices, enabling the scaling of DeltaNet similar to other linear Transformer models. The authors trained a 1.3B parameter model on 100B tokens and found that it outperforms strong linear-time baselines such as Mamba and GLA in terms of perplexity and zero-shot performance on downstream tasks.

**Strengths:**

- The paper introduces a novel hardware-efficient algorithm for training DeltaNet Linear Transformers, leveraging the WY representation of Householder matrices, which effectively addresses the parallelization limitations of previous algorithms.
- Through large-scale experiments, the authors demonstrate that DeltaNet significantly outperforms existing models like Mamba and GLA in terms of perplexity and zero-shot performance on downstream tasks.
- The new algorithm enables the scaling of DeltaNet to larger datasets and parameter sizes, which is crucial for large language models.

**Weaknesses:**

The algorithms presented in this paper are satisfactory in terms of efficiency and performance.

**Questions:**

I have no questions for this paper.

**Limitations:**

Yes

---

> ### Author Rebuttal · Authors · 2024-08-07
>
> Thanks for your review!
>
> We are adding some additional results in case they are of interest.
>
> First, we have preliminary results with 3B models trained for 167B tokens:
>
>
> | Model      | # Tokens  | wikitext PPL | arc-c | arc-e | boolq | hellaswag | openbookqa | piqa  | sciq  | winogrande | average   |
> |------------|--------|-------|-------|-------|-----------|------------|-------|-------|------------|-----------|--------------|
> | 3B-Transformer++   | 167B   | 15.39 | 29.78 | 62.96 | 63.06 | 46.15     | 27.8       | 72.96 | 91.1  | 60.62      | 56.80  |
> | 3B-DeltaNet| 167B   | 15.34 | 29.78 | 63.8  | 65.08 | 46.57     | 26.2       | 74.32 | 88.5  | 58.09      | 56.54   |
>
> We are aiming to train these models for longer (300B-1T tokens) and will report these results in the next iteration of the paper.
>
> We have also run experiments on the recently-released [MAD benchmark](https://arxiv.org/abs/2403.17844), a suite of synthetic tasks designed to test the capabilities of language models beyond perplexity. Here are the results:
>
>
>
> | Model         | Compress | Fuzzy Recall | In-Context Recall | Memorize | Noisy Recall | Selective Copy | Average |
> |---------------|----------|--------------|-------------------|----------|--------------|----------------|---------|
> | Transformer   | 51.6     | 29.8         | 94.1              | 85.2     | 86.8         | 99.6           | 74.5    |
> | Hyena         | 45.2     | 7.9          | 81.7              | 89.5     | 78.8         | 93.1           | 66.0    |
> | Multihead Hyena | 44.8   | 14.4         | 99.0              | 89.4     | 98.6         | 93.0           | 73.2    |
> | Mamba         | 52.7     | 6.7          | 90.4              | 89.5     | 90.1         | 86.3           | 69.3    |
> | GLA           | 38.8     | 6.9          | 80.8              | 63.3     | 81.6         | 88.6           | 60.0    |
> | DeltaNet      | 42.2     | 35.7         | 100               | 52.8     | 100          | 100            | 71.8    |

---

> > ### Comment · Reviewer_pqA2 · 2024-08-13
> >
> > Thanks for your response. Your responses solve my concern to some extent. I maintain my score.

---

### Official Review · Reviewer_eUyw · 2024-07-13

**Soundness:** 3
**Presentation:** 4
**Contribution:** 3
**Rating:** 7
**Confidence:** 4

**Summary:**

This paper proposes a hardware-efficient algorithm for training linear transformers with a delta update (DeltaNet; SMS21). This architecture has an attention formulation that prevents the direct application of chunk-wise parallel algorithms for computing its output. To address this issue, the authors introduce a re-parameterization of DeltaNet as a matrix-valued RNN whose recurrence is given by a generalized Householder transformation. This enables the use of WY representation which is memory efficient and eliminates the need to materialize the hidden state matrices. Experiments on synthetic benchmarks and language modeling tasks shows competitive performance compared to strong baselines (Mamba, GLA) and faster speed than the original Deltanet implementation.

**Strengths:**

- The paper is well motivated and situated with respect to prior work. It provides sufficient background for linear transformers, demonstrates great scholarship in crediting prior work, and has a clear exposition of the proposed idea. In addition, it presents an informative overview that compares the formulations of recent linear transformers that highlights their differences.
- Proposes an efficient algorithm for training linear transformers with the delta update which is a competitive variant. The re-parameterization is non-obvious and leverages WY representation for Householder matrices in a novel way. Previously, this architecture could not be easily scaled to larger models and datasets with a recurrent formulation. In addition, it introduces two competitive hybrid methods based on DeltaNet that leverage local and global full attention.
 - Demonstrates the effectiveness of the proposed approach on two synthetic benchmarks and eleven language modeling and understanding tasks compared to strong baselines such as Mamba and GLA. The results are consistent, have a good coverage, and are important for the researchers working on efficient transformers.
- The experiments are thorough and have convincing settings, namely all the variants are trained from scratch with the same configurations, there are ablations to justify the design choices, and the experimental reporting is very detailed.

**Weaknesses:**

- W1. In terms of scale, the model explores two different architectures of increasing size up to 1.3B parameters. Even though this size is considerable, it is still relatively small compared to the LLMs that are widely used such as Llama, Mistral (7B+ size). There is always the question of whether the quality is maintained with further model increase.
- W2. The improved results compared to Mamba and GLA make use of additional architectural components: convolution and local/global attention, without them the results are comparable to the other models.

**Questions:**

- Q1: What is the effect of chunk size in the chunk-wise parallel algorithm for DeltaNet? Varying the chunk size $C$ and showing its effect in efficiency would be interesting to explore.
- Q2: The chunk-level hidden states $S_{[t]}$'s are discarded to save memory. From Eq. 7, it seems that their computation depends on the previous hidden states $S_{[t-1]}$'s. Are these kept in memory for the re-computation in the backward pass?
- Q3: GLA with convolution performs worse than w/o convolution with the larger model size. Do you expect this to be the case for DeltaNet as well? It would be good to add this result if possible.

Minor:
- In Table 2, is the L1/L2 norm referring to the normalization of queries and keys? Please specify.
- In Eq.1, why is this equation showing the state $S_{[t+1]}$ instead of $S_{[t]}$? The latter is used in Eq. 2. Same for Eq. 7.
- l172: stable -> stabilize
- l213-214: we -> we follow
- l321: vallina -> vanilla

**Limitations:**

Yes, they have.

---

> ### Author Rebuttal · Authors · 2024-08-07
>
> Thanks for your review.
> ## W1: larger scale experiments
> As noted by the reviewer, it is difficult to conduct experiments at 1B+ scale. Nonetheless, we are currently running some larger-scale experiments at the 3B parameter scale. Here are some preliminary results:
>
>
> | Model      | # Tokens  | wikitext PPL | arc-c | arc-e | boolq | hellaswag | openbookqa | piqa  | sciq  | winogrande | average   |
> |------------|--------|-------|-------|-------|-----------|------------|-------|-------|------------|-----------|--------------|
> | 3B-Transformer++   | 167B   | 15.39 | 29.78 | 62.96 | 63.06 | 46.15     | 27.8       | 72.96 | 91.1  | 60.62      | 56.80  |
> | 3B-DeltaNet| 167B   | 15.34 | 29.78 | 63.8  | 65.08 | 46.57     | 26.2       | 74.32 | 88.5  | 58.09      | 56.54   |
>
> As we can see, the DeltaNet results are comparable to the Transformer++ results at this scale. We will make sure to include these results once they have finished training (we are aiming for 300B-1T tokens, depending on the availability of compute). We also plan to open-source the pretrained models so that researchers can study these models in more detail.
>
> We have also run experiments on the recently-released [MAD benchmark](https://arxiv.org/abs/2403.17844), a suite of synthetic tasks designed to test the capabilities of language models beyond perplexity. Here are the results:
>
> | Model         | Compress | Fuzzy Recall | In-Context Recall | Memorize | Noisy Recall | Selective Copy | Average |
> |---------------|----------|--------------|-------------------|----------|--------------|----------------|---------|
> | Transformer   | 51.6     | 29.8         | 94.1              | 85.2     | 86.8         | 99.6           | 74.5    |
> | Hyena         | 45.2     | 7.9          | 81.7              | 89.5     | 78.8         | 93.1           | 66.0    |
> | Multihead Hyena | 44.8   | 14.4         | 99.0              | 89.4     | 98.6         | 93.0           | 73.2    |
> | Mamba         | 52.7     | 6.7          | 90.4              | 89.5     | 90.1         | 86.3           | 69.3    |
> | GLA           | 38.8     | 6.9          | 80.8              | 63.3     | 81.6         | 88.6           | 60.0    |
> | DeltaNet      | 42.2     | 35.7         | 100               | 52.8     | 100          | 100            | 71.8    |
>
>
>
>
> ## W2: The improved results compared to Mamba and GLA make use of additional architectural components: convolution and local/global attention, without them the results are comparable to the other models.
>
> Thanks for the comment! Note that Mamba uses conv layers by default, while GLA does not. This is why we trained our own GLA+conv baselines. Our main experiments compare DeltaNet+conv against Mamba+conv and GLA+conv, i.e., we give all models the chance to use convolution layers for a fair and meaningful comparison. In this setting, DeltaNet outperforms both Mamba and GLA. Morevoer, these depthwise separable "short convolution" layers are  cheap both in terms of the number of parameters and compute, and hence we think that this convolution primitive is a practical approach to modeling local interactions.
>
>
>
>
>
>
>
>
> ## Q1 Effect of Chunk size C
> The computational complexity of the WY representation is O((N/C) * C^3) = O(NC^2). If C is too large, WY representation computation will be very expensive (recall that WY computation is fully recurrent).
>
> If the chunk size is too small, and we adapt the materialization version of FlashLinearAttention, we need to write more hidden states to HBMs, resulting in a high I/O burden and thereby slowing down the actual running speed. If we adapt the non-materialization version of FlashLinearAttention, it will lack sequence-level parallelism, which is not desirable in large-scale training. Please refer to Section 3.3 of the GLA paper for more discussions.
>
> To provide some intuition, we measured the running time (forward and backward pass) on a single A100 GPU while varying the chunk size \( C \):
>
> | Chunk Size \( C \) | Forward + Backward Time (ms) |
> |--------------------|------------------------------|
> | 16                 | 4.8738                       |
> | 32                 | 3.6616                       |
> | 64                 | 5.2382                       |
> | 128                | 16.0602                      |
>
> The experimental settings were as follows: sequence length \( \text{seq\_len} = 4096 \), batch size \( B = 2 \), head dimension \( \text{d\_head} = 128 \), model dimension \( \text{d\_model} = 2048 \), and number of heads \( \text{num\_head} = 16 \). As we can see, the moderate chunk size of 32 performs the best. When the chunk size is less than 32, the I/O cost surpasses the WY representation cost, and vice versa.
>
>
> ## Q2: Chunk state
> Yes this is correct! Concretely, we adapt the "non-materialization" version of FlashLinearAttention for chunkwise DeltaNet implementation. In this version, the hidden states of each chunk are materialized on HBMs in the forward pass, then discarded to save memory. During the backward pass, the hidden states of each chunk are recomputed.
>
>
> ## Q3: DeltaNet w/o shortconv in large scale
> Thank you for the suggestion. We agree that this will be an interesting experiment to run. We are currently using all our resources for the 3B experiments, but we will run this ablation once the 3B experiments are done.
>
> ## Minor
> Thanks for identifying the typos/errors! We will fix these.

---

> ### Comment · Reviewer_eUyw · 2024-08-12
> **Response to authors**
>
> Thank you for answering my questions and providing additional results!
>
> - The experiments with 3B model provide evidence that performance is comparable to that of a Transformer as model size increases to 3B. Given the compute constraints, my concern has been addressed adequately.
>
> - Regarding the convolutional component, I am not challenging the fairness of the comparison. The potential issue is that DeltaNet without convolution is in several tasks worse than GLA without convolution. Any insight on why that is? For completeness, I'd suggest adding some discussion about it to emphasize that it's essential to the performance of DeltaNet and reporting the results of DeltaNet w/o convolution for the 1.3B model, which are currently missing.
>
> - I found the results about the effect of chunk size insightful; it'd be good to include these results in the final version.
>
> My questions and concerns have been answered adequately except the second bullet point above. Hence, I decided to keep my original scores.

---

> ### Author Response · Authors · 2024-08-12
>
> Thanks for your feedback!
>
> ## The potential issue is that DeltaNet without convolution is in several tasks worse than GLA without convolution. Any insight on why that is?
>
> Local information plays a critical role in NLP tasks. As highlighted in [3], incorporating more local features can significantly enhance the performance of linear attention. Many recent studies also demonstrate that additional local attention mechanisms greatly improve linear attention.
>
> Short convolutions are one method to enhance local information, while the gating mechanism in GLA imposes a strong inductive bias towards local context. DeltaNet’s update rule, on its own, doesn’t prioritize local information, so without short convolutions, it may underperform in NLP tasks. However, when short convolutions are applied, DeltaNet’s ability to leverage local context is improved. Since GLA already incorporates gating mechanisms to enhance local information, short convolutions may not provide additional benefits.
>
> Additionally, it’s important to note that both the delta rule and the vanilla linear attention update rules can be seen as "context-based addressing" mechanisms. [1] emphasizes the importance of "location-based addressing," and [2] also highlights that a drawback of linear attention is its "lack of precise local token shifts and comparisons." The gating mechanism in GLA addresses this by emphasizing nearby tokens, while short convolutions can be considered another form of location-based addressing.
>
>
> - [1] https://arxiv.org/abs/1410.5401 Neural Turing Machines
>
> - [2] https://arxiv.org/abs/2402.18668 Simple linear attention language models balance the recall-throughput tradeoff
>
> - [3] https://arxiv.org/abs/2312.11135 Linear Attention via Orthogonal Memory
>
>
> ## For completeness, I'd suggest adding some discussion about it to emphasize that it's essential
>
> Thank you for your suggestion! We will include this discussion in the next iteration of the paper draft to emphasize its importance.

---

### Decision · Program_Chairs · 2024-09-25

**Decision:**

Accept (poster)

**Comment:**

It is well-known that standard softmax dense attention has quadratic complexity with respect to sequence length.  In recent years, there has been a great deal of interest in linear transformers, which have O(N) complexity, a fixed-size memory, and a theoretically unlimited context.  In practice, of course, it is not possible to fit an unlimited context into a fixed-size memory, so the "delta rule" introduces a mechanism that allows linear transformers to forget.

Unfortunately, the delta rule is not necessarily easy to implement efficiently in practice.  This paper develops an efficient parallel implementation of the delta rule, using the WY representation for computing products of Householder matrices.  The authors validate the effectiveness of the implementation with a variety of experiments, and also introduced new and compelling results on the MAD benchmark during the discussion period.

The reviewers were unanimous in recommending acceptance, and I concur.

Despite the overall high scores, my personal opinion is that the space of "alternatives to softmax attention" is at this point very crowded.  IMO, the primary reason for considering alternatives is to enable long-context.  However, the "forget" mechanism is not necessarily well-suited to long-context, and as the authors explain in the discussion, the models they are using (1-3B params) are too small for good long-context experiments.  As a result, although I believe this paper is a solid contribution, I don't recommend it for a spotlight or oral.